

# Method for processing XCP data with improved accuracy

Xinyue Zhang[1], Qisheng Zhang[1], Xiao Zhao[1], Qimao Zhang[2], Shenghui Liu[1], Shuhan Li[1], Zhenzhong Yuan[1]

[1]School of Geophysics and Information Technology, China University of Geosciences (Beijing), Beijing, 100083, China
[2]Institute of Electronics, Chinese Acadamy of Sciences, Beijing, 100190, China
*Correspondence to*: Qisheng Zhang (zqs@cugb.edu.cn)

**Abstract**. An expendable current profiler (XCP) is a device used for monitoring ocean currents. In this study, we focus on the technology available for processing XCP data and propose a more accurate method for calculating the current velocity from
the nanovolt-scale current-induced electric field measured using an XCP. In order to confirm the accuracy of the proposed data processing method, a sea test was performed in the South China Sea region, wherein, for the first time in China, ocean current/electric field data were collected from the sea surface to a depth of 1,000 m using an XCP. The current-data processing method described herein was used to determine the eastward and northward relative velocity components of the current from the measured data, which were then compared with the current data obtained using an acoustic Doppler current profiler, in
order to verify the accuracy of the measurements as well as that of the data processing method.

## 1 Introduction

Oceans cover approximately 360 million km$^2$ of the earth's surface, thus accounting for 71% of its surface area. Ocean currents play a significant role in various geological, physical, chemical, and biological processes as well as in the formation of the surrounding climates and weather patterns and the variability seen in them. Therefore, elucidating the laws that govern ocean
currents and their patterns is of great importance to the fishing and shipping industries as well as from a military point of view (Crews and Futterman, 1962).

The expendable current profiler (XCP) can gather information related to the current profile quickly (Liu and Hongkun, 2010). The XCP is jettisoned from a ship, submarine, aircraft, or similar carrying platform, so that it can rapidly measure the current and temperature profiles as it sinks and calculate the corresponding water depth based on the sinking speed (Liu et al., 2007;
Chen et al., 2011). The data obtained are transmitted to the carrying platform through a wire or by wireless communication and are processed, yielding the real-time variations in the current and temperature with the depth. XCPs can be used in oceanographic surveying, marine environmental forecasting, scientific research, and military applications and are advanced and highly efficient devices for data collection (Liu and Hongkun, 2010; Chen et al., 2010). The XCP is used for monitoring ocean currents, and an XCP was recently developed in China specifically for obtaining current-related data. The XCP works
on the following principle: the horizontal motion of the current makes it cut the vertical component of the geomagnetic field, inducing an electric field (Sanford., 1985; Zhang et al., 2008; Lin et al., 2003). In a stable geomagnetic field, the magnitude of





the induced electric field depends on the velocity of the current (Sanford, 1985; Wang et al., 2005; Yan et al., 2011). Therefore, the motion properties of the current can be determined by measuring the electric field generated (Chave et al., 1990; Wu and Chen, 1991; Shimizu and Utada, 2015). Within a certain temporal and spatial range, the current-induced electric field is a weak

direct current (DC) signal (of the order of tens of nanovolts). However, in this case, measuring the nanovolt-scale DC signal becomes a significant challenge. During the measurements corresponding to the sinking process, the above-mentioned XCP probe modulates the nanovolt-scale DC signal into an approximate single-frequency alternating current (AC) signal through mechanical rotation. However, determining the amplitude and phase of such weak single-frequency AC signals is also a technical challenge, making it difficult to process the current data. In order to solve this problem, the XCP uses a voltage-to-

frequency converter to determine the in-phase component, quadrature component, and baseline data related to the compass coil, along with the induced electric field signal. In this study, it was assumed that this single-frequency AC signal (coil and induced electric-field signal) is a modulated signal model which the amplitude of the carrier signal varies with, and a method for processing the current data to calculate the amplitude and phase of the modulated signal was used, in order to determine the eastward and northward relative velocity components of the current.

In this study, we propose a method for processing XCP current data in order to improve its accuracy. To efficiently calculate the current parameters based on XCP data, two steps are essential. The first is to calculate the electric field that is generated by a given current and use the results of this calculation to determine the speed and direction of the current corresponding to the measured electric field. The second is to determine the effect of placing the probe in seawater on the electric field, in order to ensure the accuracy of the current measurements. In addition, owing to the differences in the microchips, capacitors, and

resistors used in different XCP probes, the magnitude of the simulated signal may deviate from the theoretical value; this will directly affect the results of the subsequent data processing. Thus, in order to ensure the accuracy of the XCP data processing, the XCP probe used in this study was calibrated. Finally, the accuracy of the method used for processing the current data was confirmed through a sea test.

## 2. Processing of ocean current data obtained using XCP

### 2.1 Principle underlying XCP ocean current data processing

In a rectangular coordinate system consisting of the x-axis (east), y-axis (north), and z-axis (vertical), an ocean current with velocity $V$ flows horizontally in any direction, and the measurement direction angle between the induced voltage and the y-axis is $\theta$. The electromotive force, $\Delta\Phi_1$, induced by the ocean current, as measured between two points on horizontally placed electrodes and separated by distance $L$, is given by the following equation (Liu and Hongkun, 2010; Sanford, 1971):

$$\Delta\Phi_1 = F_z \times (\bar{V} - \bar{\bar{V}}) \bullet \bar{L}$$
$$= (V_E - \overline{V_E})F_z L \cos\theta - (V_N - \overline{V_N})F_z L \sin\theta$$
(1)

where $V_E$, $\overline{V_E}$, $V_N$, and $\overline{V_N}$ represent the eastward component of the current velocity, average velocity of the eastward



component, northward component of the velocity, and average velocity of the northward component, respectively, and $F_Z$ is

the vertical component of the geomagnetic field. Equation (1) describes the electric field induced by an ocean current. From

the equation, it can be seen that the measured voltage is not only proportional to the distance L measured by the probe but also

to $(\vec{V} - \overline{\overline{V}})$, the relative velocity of the ocean current. Therefore, $(\vec{V} - \overline{\overline{V}})$ can be obtained by measuring $\Delta\Phi_1$. However,

because it is difficult to determine $\overline{\overline{V}}$ in practice, the measured data for the electromagnetic fields induced by ocean currents

are used to calculate the relative velocity of the currents.

Based on the model proposed by Sanford et al. (Sanford, 1971), Eq. (1) can be improved by adding a compensation factor for

the XCP probe, as shown in Eq. (2), where K is indicative of the impact of the XCP probe on the distribution of the current-

induced electric field around the probe when it is placed in seawater. Numerical simulations, forward modelling, and physical

simulations have shown that, when the XCP probe is placed in the test area, there is an approximately two-fold increase in the

strength of the current-induced electric field as measured by the electric-field sensor of the XCP. Thus, the value of K is

normally taken to be 1.

The electric field, $\psi_2$, generated by the two electrodes cutting through the geomagnetic field during the descent of the XCP

probe is expressed by Eq. (3), where $F_H$ is the horizontal geomagnetic field, W is the descent velocity of the XCP probe, L

is the interelectrode spacing, and $\theta$ is the angle between the measurement electrodes and the y-axis (magnetic north).

Therefore, the total voltage, $\Delta U_e$, acting on the electric-field sensor of the XCP is given by Eq. (4).

$$\Delta\Phi_1 = (1+K)[(V_E - \overline{V_E})F_z L\cos\theta - (V_N - \overline{V_N})F_z L\sin\theta] \tag{2}$$

$$\psi_2 = F_H L W \sin\theta \tag{3}$$

$$\begin{aligned}\Delta U_e &= \Delta\Phi_1 + \psi_2 \\ &= F_z L(V_E - \overline{V_E})(1+K)\cos\theta - [F_z(V_N - \overline{V_N})(1+K) - F_H W]L\sin\theta\end{aligned} \tag{4}$$


$\Delta U_e$ and $\psi_2$, namely, the voltage signal measured by the electrodes and the coil signal, are converted into pulse signals by

a voltage/frequency converter; the magnitude of the voltage is represented by the magnitude of the frequency of the signal

(Zhang et al., 2011). During the measurement process, the modulated signals are demodulated by the probe using electrical

circuits, based on the compass coil signals, and the demodulated signals are transmitted to the surface XCP float (Gandolfi et

al., 1972). They are then relayed to and stored in the wireless ocean current data receiver in the deck unit.

The modulated signals can be modeled as follows:

$$F(t) = A\cos(\omega t + \varphi) + C + Dt + Et^2 + \delta \tag{5}$$

where ω is the spinning angular frequency of the probe; φ is its phase position; C, D, and E are the delay coefficients of the

circuits; and δ is the measurement noise.


In fact, the primary data recorded by the measurement instruments include the period counts of the useful signals and the coil

signals, and the two types of signals are demodulated into their in-phase component $I_n$, quadrature component $Q_n$, and

baseline component, $B_n$. The measurement of the period of the voltage/frequency converter of F(t) is shown in Figure 1,

where CIR represents the control signals of the in-phase component counter and CQR represents the control signals of the

quadrature component counter. According to Figure 1, the relationship between $I_n$, $Q_n$, and $B_n$ of the modulated signals is

as follows:

$$I_n = \int_{t0}^{t2} F(t)dt - \int_{t2}^{t4} F(t)dt \tag{6}$$

$$Q_n = \int_{t1}^{t3} F(t)dt - \int_{t3}^{t5} F(t)dt \tag{7}$$

$$B_n = \int_{t0}^{t4} F(t)dt \tag{8}$$

and $t0 = 0, t1 = \dfrac{T_{n-1}}{4}, t2 = \dfrac{T_{n-1}}{2}, t3 = \dfrac{3T_{n-1}}{4}, t4 = T_n, t5 = \dfrac{5T_n}{4}$. Here, $T_{n-1}$ and $T_n$ are the (n-1)th and nth periods of rotation

of the XCP probe as it is going down.

As can be seen from Figure 1, $\begin{aligned} T_{In} &= T_{Bn} = T_n, T_{Qn} = \dfrac{5T_n}{4} - \dfrac{T_{n-1}}{4} \\ \omega t_i &= \dfrac{\pi}{2} i, \quad i = 1, \dots, 5 \end{aligned}$. Here, $T_{In}$, $T_{Qn}$, and $T_{Bn}$ are the periods of the in-phase

component, quadrature component, and baseline component. Substituting Eq. (5) in Eq. (6), (7), and (8) results in Eq. (9),
(10), and (11).

$$I_n = -\frac{4A}{2\pi}T_{In}\sin\phi + C(T_{n-1} - T_n) + \frac{D}{2}\left(\frac{T_{n-1}^2}{2} - T_n^2\right) + \frac{E}{3}\left(\frac{T_{n-1}^3}{4} - T_n^3\right) \tag{9}$$

$$Q_n = -\frac{4A}{2\pi}T_{Qn}\cos\phi + \frac{5C}{4}(T_{n-1} - T_n) + \frac{D}{32}(17T_{n-1}^2 - 25T_n^2) + \frac{E}{192}(53T_{n-1}^3 - 125T_n^3) \tag{10}$$

$$B_n = CT_n + \frac{D}{2}T_n^2 + \frac{E}{3}T_n^3 \tag{11}$$

C, D, and E can be obtained by solving Eq. (11) for three adjacent periods, that is, for $B_{n-1}$, $B_n$, and $B_{n+1}$. By making the

circuit delay correction for Eq. (9) and (10) based on Eq. (11), $I_n^{'}$ and $Q_n^{'}$, which are given by Eq. (12) and (13), can be

determined:

$$I_n^{'} = -\frac{4A}{2\pi}T_{In}\sin\phi = I_n - [C(T_{n-1} - T_n) + \frac{D}{2}\left(\frac{T_{n-1}^2}{2} - T_n^2\right) + \frac{E}{3}\left(\frac{T_{n-1}^3}{4} - T_n^3\right)] \tag{12}$$

$$Q_n^{'} = -\frac{4A}{2\pi}T_{Qn}\cos\phi = Q_n - [\frac{5C}{4}(T_{n-1} - T_n) + \frac{D}{32}(17T_{n-1}^2 - 25T_n^2) + \frac{E}{192}(53T_{n-1}^3 - 125T_n^3)] \tag{13}$$





The two components of the modulated signals, $F_I$ and $F_Q$, can be calculated based on $I'_n$ and $Q'_n$ as follows:

$$F_I = -A\sin\varphi = \frac{2\pi}{4T_{In}}I'_n \quad F_Q = -A\cos\varphi = \frac{2\pi}{4T_{Qn}}Q'_n \tag{14}$$

Then, based on $F_I$ and $F_Q$, one can determine the amplitude, $A_F$, and phase, $\varphi_F$, of the modulated signals:

$$A_F = \sqrt{F_I^2 + F_Q^2} \quad \varphi_F = tg^{-1}(\frac{F_I}{F_Q}) \tag{15}$$

Next, from the gain recovery of the instrument, one can obtain the amplitudes and phases of the useful voltage and coil voltage signals:

$$A_E \angle \varphi_E \tag{16}$$

$\quad A_C \angle \varphi_C$ \hfill (17)

Finally, the eastward and northward relative velocity components of the current, represented by $V_{Er}$ and $V_{Nr}$, respectively, can be calculated using Eq. (16) and (17).

$$V_{Er} = (V_E - \overline{V_E}) = \frac{A_E}{F_z L(1+K)}\cos\psi \tag{18}$$

$$V_{Nr} = (V_N - \overline{V_N}) = \frac{A_E}{F_z L(1+K)}\sin\psi + W\frac{F_H}{F_z(1+K)} \tag{19}$$

where $\psi = \frac{3}{2}\pi + \varphi_C - \varphi_E$ 。

## 2.2 Procedure for processing XCP ocean current data

The procedure for processing the XCP current data is as follows:

1. Based on three adjacent periods, $B_{n-1}$, $B_n$, and $B_{n+1}$, determined from the log of the measuring instrument, solve the simultaneous equations shown below and calculate C, D, and E.

$\quad B_{n-1} = CT_{n-1} + 1/2\,DT_{n-1}^2 + 1/3\,ET_{n-1}^3$ \hfill (20)

$\quad B_n = CT_n + 1/2\,DT_n^2 + 1/3\,ET_n^3$ \hfill (21)

$\quad B_{n+1} = CT_{n+1} + 1/2\,DT_{n+1}^2 + 1/3\,ET_{n+1}^3$ \hfill (22)

2. Calculate the periods of the in-phase component, quadrature component, and baseline component, namely, $T_{In}$, $T_{Qn}$, and $T_{Bn}$, respectively.

$\quad T_{In} = T_n, \; T_{Qn} = 5/4\,T_n - 1/4\,T_{n-1,} \; T_{Bn} = T_n$ \hfill (23)

3. Determine $F_I$ and $F_Q$:



$$I_n = -\frac{4A}{2\pi} T_{In} \sin\phi + C(T_{n-1} - T_n) + D(1/2 T_{n-1}^2 - T_n^2)/2 + E(1/4 T_{n-1}^3 - T_n^3)/3 \tag{24}$$

$$Q_n = -\frac{4A}{2\pi} T_{Qn} \cos\phi + C(T_{n-1} - T_n)5/4 + D(17 T_{n-1}^2 - 25 T_n^2)/32 + E(53 T_{n-1}^3 - 125 T_n^3)/192 \tag{25}$$

Set $I_n' = -\dfrac{4A T_{In}}{2\pi} \sin\phi$. Then, on combining Eq. (26) with (24), you get

$$I_n' = -\frac{4A T_{In}}{2\pi} \sin\phi = I_n - \left[ C(T_{n-1} - T_n) + D(1/2 T_{n-1}^2 - T_n^2)/2 + E(1/4 T_{n-1}^3 - T_n^3)/3 \right] \tag{26}$$

Set $Q_n' = -\dfrac{4A T_{Qn}}{2\pi} \cos\phi$, and substitute Eq. (25) into Eq. (27),

$$Q_n' = -\frac{4A T_{Qn}}{2\pi} \cos\phi$$
$$= Q_n - \left[ C(T_{n-1} - T_n)5/4 + D(17 T_{n-1}^2 - 25 T_n^2)/32 + E(53 T_{n-1}^3 - 125 T_n^3)/192 \right] \tag{27}$$

Based on $I_n'$ and $Q_n'$, the two components of the modulated signals $F_I$ and $F_Q$ can be calculated as follows:

$$F_I = -A\sin\phi = \frac{2\pi}{4 T_{In}} I_n'$$
$$= \frac{\pi}{2 T_{In}} \left\{ I_n - \left[ C(T_{n-1} - T_n) + D(1/2 T_{n-1}^2 - T_n^2)/2 + E(1/4 T_{n-1}^3 - T_n^3)/3 \right] \right\} \tag{28}$$

$$F_Q = -A\cos\phi = \frac{2\pi}{4 T_{Qn}} Q_n'$$
$$= \frac{\pi}{2 T_{Qn}} \left\{ Q_n - \left[ C(T_{n-1} - T_n)5/4 + D(17 T_{n-1}^2 - 25 T_n^2)/32 + E(53 T_{n-1}^3 - 125 T_n^3)/192 \right] \right\} \tag{29}$$

4. Next, filter $F_I$ and $F_Q$ in order to get $\overline{F}_I$ and $\overline{F}_Q$ (this is done using a Bartlett (i.e., a triangle window) whose weight is $W_n$) (Zoltan, 2012):

$$\overline{F}_I = \frac{2\pi}{4} \sum_{n=1}^{Nav} W_n I_n' / T_{In} = \sum_{n=1}^{Nav} W_n F_{In} \tag{30}$$

$$\overline{F}_Q = \frac{2\pi}{4} \sum_{n=1}^{Nav} W_n Q_n' / T_{Qn} = \sum_{n=1}^{Nav} W_n F_{Qn} \tag{31}$$

5. Based on the two components calculated above, the amplitude, $A_F$, and phase, $\varphi_F$, of the modulated signals can be obtained:

$$A_F = \left( \overline{F}_I^2 + \overline{F}_Q^2 \right)^{1/2}, \quad \varphi_F = \tan^{-1} \frac{-\overline{F}_I}{-\overline{F}_Q}, \quad F = A_F \angle \varphi_F \tag{32}$$

6. The amplitude and phase of the useful voltage signal and coil voltage signal are calculated using the procedure described in steps 3–5. It can be concluded that the amplitude and phase of the electric-field-related current signal and coil signal are $F_E = A_E \angle \varphi_E$ and $F_C = A_C \angle \varphi_C$, respectively.



7.   The circuit diagram of the hardware used for data processing is shown in Figure 2. One can determine the original

electric field and coil electric field as follows:

$$E_E = \frac{F_E}{G_{EFVF}G_{EF2}} - E_C \frac{G_{COR2}}{G_{EF2}}, \ E_C = \frac{F_C}{G_{CCVF}G_{CC2}} \tag{33}$$

where   $G_{EF2} = \frac{E_7}{E_E}$,   $G_{COR2} = \frac{E_7}{E_C}$,   $G_{CC2} = \frac{E_8}{E_C}$,   $G_{EFVF} = \frac{F_E}{E_7}(Hz/V)$, and   $G_{CCVF} = \frac{F_C}{E_8}(Hz/V)$.

8.   Calculate the northward speed and the eastward speed:

$$u = \frac{A_E}{F_Z L(1+K)}\cos\psi \tag{34}$$

$$v = -\frac{A_E}{F_Z L(1+K)}\sin\psi + W\frac{F_H}{F_Z(1+K)} \tag{35}$$

where   $\psi = \frac{3}{2}\pi + \varphi_C - \varphi_E$.

Based on the velocity model, which uses   $W = -2*a*t - b$,

the relation between the depth of the XCP probe in the ocean and time is   $Z = -1*(c+b*t+a*t^2)$

where $a = -3.5*10^\wedge(-4); b = 3.85; c = 3.1$. This is the experience value of the development of XCP probe.

## 3 Results and discussion

Because laboratory simulations of the marine environment are difficult to perform, in order to verify the accuracy of the

proposed method for processing the XCP current data, we performed a sea test using the XCP. An XCP probe (probe 1) was

combined with an XCP buoy and released in the test area. The location of the release point was 21°59.420'N, 118°10.310'E,

and data were collected up to a maximum depth of 1,151 m. The changes in the rotational frequency and coil signal (Ec) of

the probe during sinking are shown in Figure 3. The collected Ec data were processed to obtain the ocean current velocity data,

which were then combined with the collected temperature data to plot the curves shown in Figure 2.

The data shown in Figures 4 and 5 were collected at a location 44.3 ° east by south of Shantou, Guangdong Province, China

and approximately 215 km away from Shantou, Guangdong Province, China. For comparison, Figure 5 shows the ocean

current velocity data as measured by an acoustic Doppler current profiler (ADCP) (Zoltan, 2012). The ADCP used was an OS-

75K ADCP supplied by RDI and had a maximum profiling depth of 700 m (Liu, 2016). A comparison of Figures 4 and 5

shows that the ocean current velocity as measured by the XCP was generally consistent with that measured by the ADCP.

From 16 m below sea level(BSL) to 600 m BSL, the velocity of the eastward ocean current changed gradually from -0.6m s$^{-1}$



to -0.1 m s$^{-1}$, while the velocity of the northward ocean current changed gradually from -0.5 m s$^{-1}$ to -0.96 m s$^{-1}$. The current at the top layer ran southwestward in a direction approximately parallel to the coastal line in Guangdong Province, China.

## 4 Conclusions

Based on the theoretical principles underlying the processing of current data obtained using an XCP, in this study, we developed a method for processing XCP data in order to improve the accuracy of the measurements. In addition, a sea test was performed to evaluate the accuracy of the proposed method. Relationships derived based on theoretical studies were used to process the data collected from the sea test and to plot the current velocity data. Moreover, the current velocity data obtained using the XCP were compared with those obtained using an ADCP. It was found that the trends in these two sets of velocity
data were essentially consistent, confirming the accuracy of the proposed method for processing XCP current data.

## 5 Acknowledgements

This work was supported by the Fundamental Research Funds for the Central Universities of China (No. 2652014070), the National Natural Science Foundation of China (No. 41574131), and the National "863" Program of China (No. 2012AA061102).

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



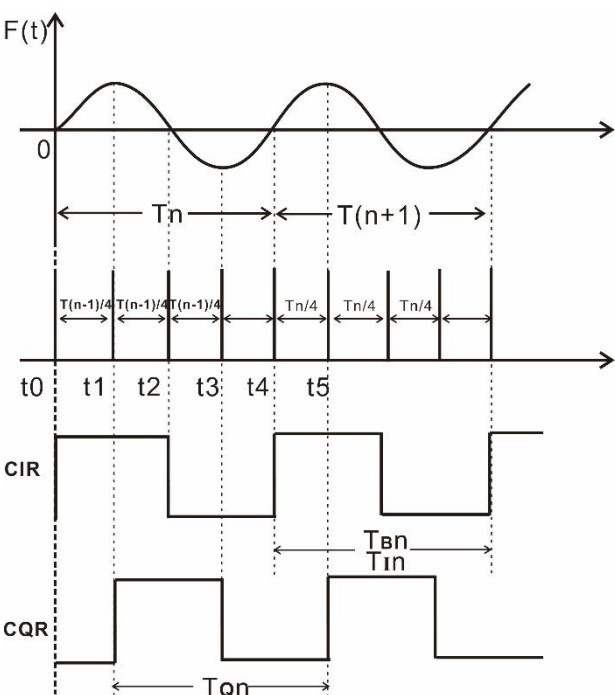

**Figure 1: Measurement of period of voltage/frequency converter of F(t).**

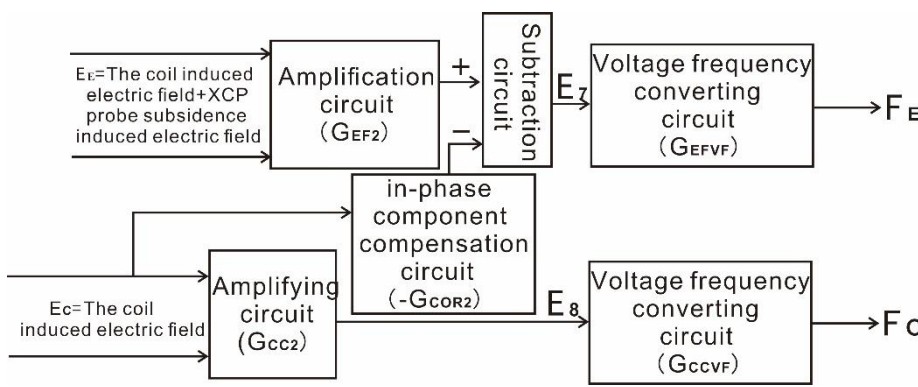


**Figure 2: Circuit diagram of hardware used for data processing.**



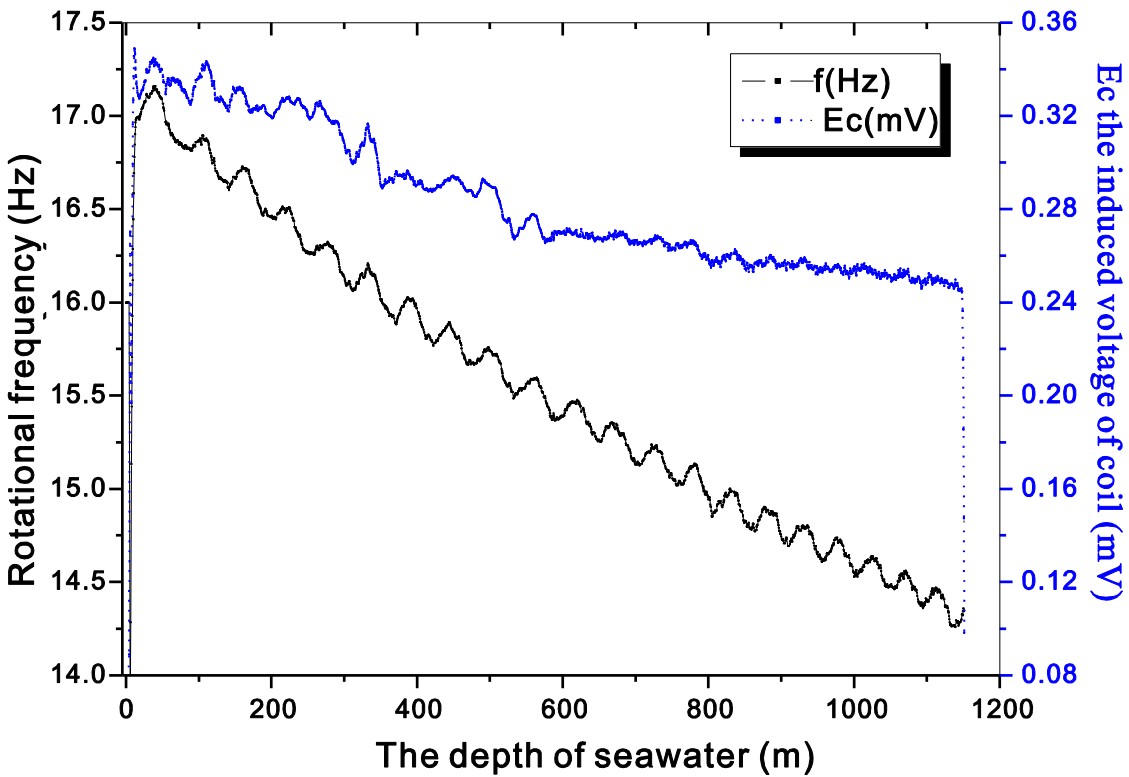

**Figure 3 Rotational frequency of XCP probe and change in amplitude of coil signal, $E_C$ .**


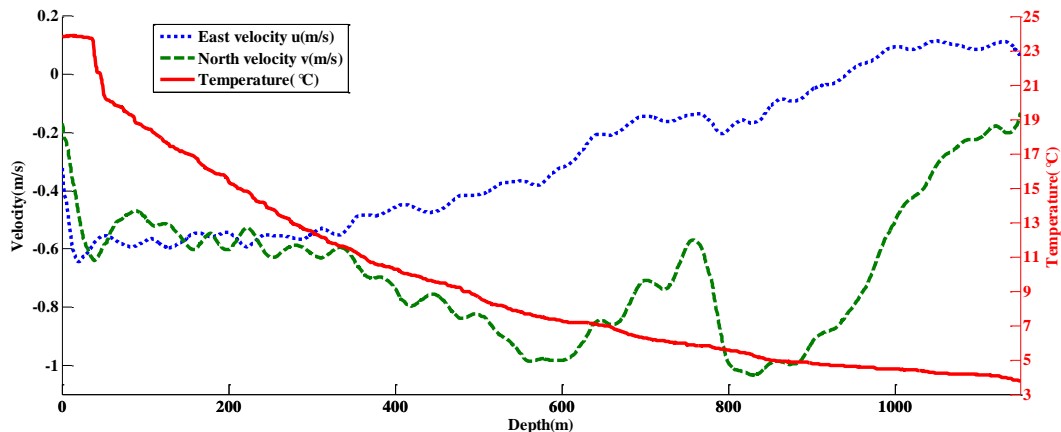

**Figure 4: Ocean current velocity and temperature data as obtained using XCP probe .**





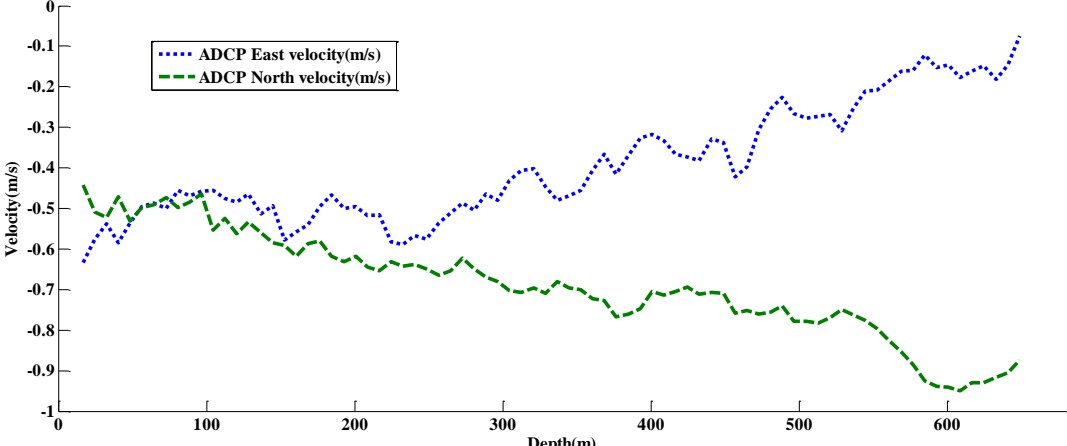

**Figure 5: Ocean current velocity data (N 21°59.42′, E 118°10.31′) as obtained using ADCP.**
