# Peer review of "Method for processing XCP data with improved accuracy"

_Geoscientific Instrumentation, Methods and Data Systems, 2016_

## Referee Comment (RC1) · Anonymous Referee #2 · 10 Feb 2017

First of all, this paper addresses relevant scientific questions within the scope of GI and presents novel concepts and ideas about the method for processing XCP data. The title clearly reflects the contents of the paper and the abstract provides a concise and complete summary. This paper outlines the scientific methods and assumptions clearly, and then it reaches substantial conclusions, the results in discussion part are sufficient to support the interpretations and conclusions. The calculations are sufficiently complete and precise. The number and quality of references are appropriate. The overall presentation is well structured and clear. But I still have some questions, so minor revisions should be made. 1. Line35: The unit of electric field strength is V/m, so what's the meaning of 'nanovolts' you mentioned here? 2. Line46: Two essential steps are presented here, while the second one is scarcely mentioned in the text, with only one result. 3. Line57: In the text you mention 'the induced voltage direction', but

the voltage is scalar and has no direction. 4. Line61: The meaning of the 'average velocity' is unclear, please make a clear definition. After reading through the full paper, I find the language is fluent and precise.

Please also note the supplement to this comment:
http://www.geosci-instrum-method-data-syst-discuss.net/gi-2016-43/gi-2016-43-RC1-supplement.pdf

---

## Referee Comment (RC2) · Anonymous Referee #1 · 13 Feb 2017

My recommendation is to accept the contribution with the technical corrections addressed by referee #2.

---

## Author Comment (AC1) · 14 Feb 2017

To esteemed anonymous referee #2. Your comments impress us a lot. We really appreciate your time and energy for reviewing our manuscript. It is hard for us to express our grateful feeling. In fact, we have learned many things during this revision process, and such experience would be very helpful for our future study. We herewith provide our response to your comments as below: 1. Line35: The unit of electric field strength is V/m, so what's the meaning of 'nanovolts' you mentioned here? Our response: The 'nanovolts' refers to the induced electromotive force on the electrode.

2. Line46: Two essential steps are presented here, while the second one is scarcely mentioned in the text, with only one result. Our response: This paper focuses on the method for processing XCP data, so we mainly worked on theoretical research

rather than experimental verification. We also made physical implementation and wrote another paper which has been published in Mathematical Problems in Engineering. Because we used an experimental result in the process of formula derivation, so we quoted the following related references in the paper.

Zhang, Q. S., Xiao, Z., Xinyue, Z., et al.: Influence of Expendable Current Profiler Probe on Induced Electric Field of Ocean Currents, Mathematical Problems in Engineering. 2016, Article ID 9812929, 9 pages, 2016.

3. Line57: In the text you mention 'the induced voltage direction', but the voltage is scalar and has no direction. Our response: Thanks for your suggestion, we refer to the direction of the induced electric field.

4. Line61: The meaning of the 'average velocity' is unclear, please make a clear definition. Our response: The sea has a certain depth, and different current layer with different velocity, in order to ensure the accuracy of calculation results, we define the 'average velocity' is the average of current velocity in different current layer.

Please also note the supplement to this comment:
http://www.geosci-instrum-method-data-syst-discuss.net/gi-2016-43/gi-2016-43-AC1-supplement.pdf

---

## Author Comment (AC2) · 14 Feb 2017

To esteemed anonymous referee #1

We have made the technical corrections addressed by referee #2. We really appreciate your time and energy for reviewing our manuscript. It is hard for us to express our grateful feeling. In fact, we have learned many things during this revision process, and such experience would be very helpful for our future study.